# Examination of the Hydrogen Incorporation into Radio Frequency-Sputtered Hydrogenated SiN$_x$ Thin Films

**Nikolett Hegedüs [1,2,3], Riku Lovics [1], Miklós Serényi [1], Zsolt Zolnai [1], Péter Petrik [1], Judit Mihály [4], Zsolt Fogarassy [1], Csaba Balázsi [1] and Katalin Balázsi [1,*]**

1 Centre for Energy Research, Institute for Technical Physics and Materials Science, Konkoly-Thege M. Str. 29-33, 1121 Budapest, Hungary; nhegedus@guardian.com (N.H.); drlovicsriku@gmail.com (R.L.); serenyi.miklos@energia.mta.hu (M.S.); zolnai.zsolt@ek-cer.hu (Z.Z.); petrik.peter@ek-cer.hu (P.P.); fogarassy.zsolt@ek-cer.hu (Z.F.); balazsi.csaba@ek-cer.hu (C.B.)
2 Doctoral School of Materials Science and Technologies, Óbuda University, Bécsi Str. 96/B, 1030 Budapest, Hungary
3 Guardian Orosháza Ltd., Csorvási u. 31, 5900 Orosháza, Hungary
4 Research Centre for Natural Sciences, Institute of Materials and Environmental Chemistry, Magyar Tudósok Krt. 2, 1117 Budapest, Hungary; mihaly.judit@ttk.mta.hu
* Correspondence: balazsi.katalin@ek-cer.hu

**Abstract:** In this work, amorphous hydrogen-free silicon nitride (a-SiN$_x$) and amorphous hydrogenated silicon nitride (a-SiN$_x$:H) films were deposited by radio frequency (RF) sputtering applying various amounts of hydrogen gas. Structural and optical properties were investigated as a function of hydrogen concentration. The refractive index of 1.96 was characteristic for hydrogen-free SiN$_x$ thin film and with increasing H$_2$ flow it decreased to 1.89. The hydrogenation during the sputtering process affected the porosity of the thin film compared with hydrogen-free SiN$_x$. A higher porosity is consistent with a lower refractive index. Fourier-transform infrared spectroscopy (FTIR) confirmed the presence of 4 at.% of bounded hydrogen, while elastic recoil detection analysis (ERDA) confirmed that 6 at.% hydrogen was incorporated during the growing mechanism. The molecular form of hydrogen was released at a temperature of ~65 °C from the film after annealing, while the blisters with 100 nm diameter were created on the thin film surface. The low activation energy deduced from the Arrhenius method indicated the diffusion of hydrogen molecules.

**Keywords:** SiN$_x$:H; refractive index; activation energy; structure

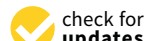



## 1. Introduction

Hydrogenated silicon nitride (SiN$_x$:H) films are widely used in the microelectronics industry to enhance the efficiency of silicon-based light emitters [1] or to improve the efficiency of silicon solar cells as the antireflective and passivation layer on the front surface of such device structures. Silicon nitride (Si$_3$N$_4$, hereinafter referred to as SiN) thin films may be applied as inorganic gate insulators in organic thin film transistors (OTFTs) [2]. SiN:H is also an appropriate material for charge trap functional region of non-volatile memory (NVM) structures [3]. Their tunable refractive index together with the low extinction coefficient enable the application of SiN$_x$ as an excellent antireflection thin film [4]. Furthermore, SiN$_x$:H containing multilayer stacks with a gradient refractive index profile were recommended to further decrease the optical losses [5]. Apart from the optical properties, the passivation effect of SiN$_x$:H layers is also substantial since recombination losses significantly restrict the performance of solar cells. Hydrogenated silicon nitride films have been reported to show a good surface and bulk passivation effect after annealing due to atomic hydrogen diffusion to the surface [6]. As dangling bonds are healed by hydrogen, the number of recombination centers can be reduced, which results in increasing carrier lifetime. However, the molecular hydrogen migration requires a higher activation energy and low diffusivity [7].

The most common techniques for deposition of the silicon nitride films with or without hydrogen addition are different types of chemical vapor deposition (CVD), such as plasma-enhanced chemical vapor deposition (PECVD) [8,9], remote plasma-enhanced CVD (RPECVD) [10], electron cyclotron resonance (ECR) [11,12], hot-wire CVD (HWCVD) [13], and extended thermal plasma CVD (ETPCVD) [14]. CVD-deposited film always contains hydrogen but its amount cannot be directly controlled during the preparation process, only by several deposition parameters, such as the ratio of precursor gases or the substrate temperature [9,12]. Due to this fact, the magnetron sputtering technique could be the alternative fabrication method for directly controlled hydrogen concentration via adjusting the applied hydrogen gas flow to the chamber [15].

Direct current (DC) magnetron sputtering [16], radio frequency (RF) sputtering [17], and high-power impulse magnetron sputtering (HiPIMS) [18] were also proved to be proper methods to produce $SiN_x$:H thin films at a lower substrate temperature. In the case of different sputtering techniques, it is possible to directly control the amount of hydrogen by adjusting the applied hydrogen gas flow. K. Mokeddem et al. developed the DC magnetron sputtering (DCMS) technique to prepare hydrogenated amorphous silicon nitride thin films in argon gas flow mixed with molecular hydrogen and nitrogen [16]. These films presented a large band gap and showed a nearly stoichiometric composition, and both nitrogen and hydrogen were incorporated into the structure. F.L. Martinez et al. used electron cyclotron resonance plasma-enhanced CVD for amorphous hydrogenated silicon nitride (a-$SiN_x$:H) film deposition under different values of gas flow ratio, deposition temperature, and microwave power [12]. The formation of Si–H and N–H as competitive processes occurred during the film growth. They showed that the substitution of N–H bonds with Si–H bonds was driven by the tendency for chemical order or maximum bonding energy. V. Tiron et al. used the reactive HiPIMS technique to fabricate $SiN_x$:H thin films. Their coating showed a very smooth surface with a dense homogeneous amorphous and amorphous to nanocrystalline structure [18]. This coating revealed a diffusion process of atomic H into the Si substrate, indicating the presence of numerous hydrogen bonds (Si–H and N–H) that could passivate structural defects and reduce the number of recombination centers in silicon bulk.

The physical, electrical, and optical properties of amorphous $SiN_x$ and $SiN_x$:H thin films strongly depend on film composition and the applied deposition technique. Ellipsometry is a widely used non-destructive technique for the optical characterization of a wide range of thin layer-on-substrate material systems. The optical properties cannot be deduced directly from the raw measurement data but indirectly from the ellipsometric modeling process. Therefore, the choice of the applied ellipsometric model is an important point of data evaluation. Boulesbaa et al. developed an effective medium approximation (EMA) model, which considers $SiN_x$:H films as a mixture of silicon (Si), stoichiometric silicon nitride, and hydrogen ($H_2$) [19]. In addition to experimental work, the optical properties of SiN:H films have also been studied by theoretical models. F. de Brito Mota et al. developed an interatomic potential to describe a-$SiN_x$:H thin films of varying nitrogen content [20]. They found that hydrogen incorporation into silicon nitride films leads to the reduction of dangling bonds corresponding to undercoordinated silicon and nitrogen atoms. Tao et al. calculated the optical properties of $SiN_x$:H films based on the density functional theory [21]. They found that the hydrogen incorporation into the silicon nitride films had a healing effect by saturating the dangling bonds, which leads to the decrease of absorption coefficient and refractive index of the films.

Different deposition conditions may result in modified coating structure and material properties. In this work, amorphous hydrogen-free (a-$SiN_x$) and a-$SiN_x$:H thin films were deposited by RF sputtering onto two kinds of substrates (Si (001) and glass) with various amounts (0–12 sccm) of hydrogen. The effects of the hydrogen flow on the optical and structural properties were investigated.

## 2. Experimental Section

Both types of silicon nitride thin films (a-SiN$_x$, a-SiN$_x$:H) were deposited by a Leybold Z400 Radio Frequency Sputtering (RFS) tool (Figure 1). The base pressure was $2 \times 10^{-5}$ mbar. A circular Si target with a diameter of ~76 mm (Kurt J. Lesker Comp., undoped with 99.99% purity) and N$_2$ gas source were applied for deposition of the hydrogen-free silicon nitride films.

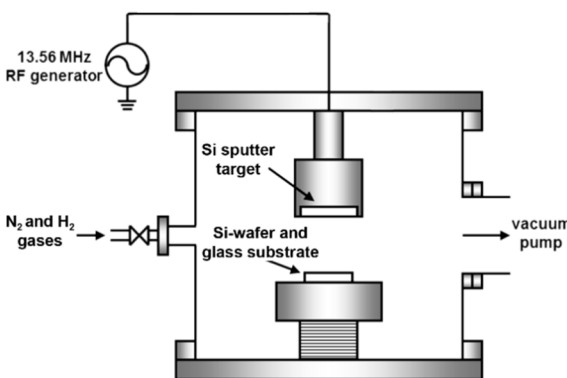

**Figure 1.** Schematic view of the sputtering system.

In the case of SiN$_x$:H films, N$_2$ and hydrogen with a flow rate in the range from 0 to 12 sccm were added. The gas flow rates were controlled by adjustable mass flow controllers (MFCs). The sputtering parameters are summarized in Table 1.

**Table 1.** Summary of sputtering parameters (U = 2 kV, p$_{total}$ = 2.5 × 10$^{-2}$ mbar for all thin films. Single-side polished (SSP), double-side polished (DSP). * fully closed valve.

| Nr. | Thin Film | $p_{H2}$ ($10^{-4}$ mbar) | $p_{H2}$ % of Total Pressure (%) | Sputtering Time (min) | Flow (sccm) | Substrate |
|---|---|---|---|---|---|---|
| **R1** | a-SiN$_x$ | 0 | 0 | 30 | 0 * | SSP |
| **R2** | a-SiN$_x$ | 0 | 0 | 80 | 0 * | SSP, DSP glass |
| **S1** | a-SiN$_x$:H | 0.5 | 0.2 | 30 | 0.9 | SSP |
| **S2** | a-SiN$_x$:H | 0.8 | 0.32 | 30 | 1.6 | SSP |
| **S3** | a-SiN$_x$:H | 1.5 | 0.6 | 30 | 3 | SSP |
| **S4** | a-SiN$_x$:H | 3.3 | 1.32 | 30 | 6 | SSP |
| **S5** | a-SiN$_x$:H | 7.9 | 3.16 | 30 | 12 | SSP |
| **S6** | a-SiN$_x$:H | 3.3 | 1.32 | 80 | 12 | SSP, DSP glass |

Single-side and double-side polished (SSP and DSP, respectively) intrinsic un-doped crystalline (001) Si wafers and ~40 mm × 40 mm size soda lime glass slides with nominal thickness of 4 mm (Guardian Orosháza Ltd., Orosháza, Hungary) were used as substrates. The target–substrate distance was constantly kept as 50 mm. The sputtering process was applied at room temperature. The thin film properties are not directly determined by the process parameters because the plasma diagnostic results are not available. Fourier-transform infrared spectroscopy (FTIR) was used for investigating the hydrogen bond configuration. FTIR measurements were performed by a Varian 7000 FTIR spectrometer (Agilent Technologies, Santa Clara, CA, USA) connected to a UMA 600 IR microscope equipped with a mercury–cadmium–telluride (MCT) detector. The absorbance spectra were recorded in the wavenumber range between 600 and 4000 cm$^{-1}$ with spectral resolution of 4 cm$^{-1}$. The evaluation of the measured spectra including the baseline correction by adjusted polynomial fit was done using Origin 2019b (64-bit) software, version 9.6.5.169. Transmission electron microscopy (TEM, Philips CM20 with 200kV accelerating voltage, Hillsboro, OR, USA) and Cs-corrected (S)TEM (FEI Themis 200 with accelerating voltage 200 kV) were applied for the structural characterization of the thin films. TEM samples were prepared by conventional Ar ion milling technique. Scanning electron microscopy

(SEM, LEO 1540 XB with accelerating voltage 5 kV) was used to investigate the morphology of the thin films' surfaces before and after annealing. SE measurements were performed by a Woollam M2000DI UV–VIS ellipsometer (Woollam Co., Lincoln, NE, USA) at the angles of 70° and 75° with a compensator frequency of 20 Hz. The wavelengths used for the measurements ranged from 200 to 1600 nm. The hydrogen-free $SiN_x$ thin film was modeled by the Cauchy–Urbach equation and it was considered as a mixture of silicon (Si), hydrogen-free stoichiometric silicon nitride ($Si_3N_4$), and void. Based on this consideration, the thin films were modeled by the Bruggeman-type EMA model. All modeling and calculations were performed using Woollam VASE software (versions of 3.83 and 3.84). The activation energy of the surface modification was determined from the Arrhenius equation using an optical measurement configuration, the details of which are given elsewhere [22]. The 1.6 MeV Rutherford backscattering spectrometry/elastic recoil detection analysis (RBS/ERDA) measurements were performed in a scattering chamber with a two-axis goniometer, which was connected to the 5 MV EG-2R Van de Graaff accelerator operated at the Wigner Research Center of Physics in Budapest, Hungary. The $4He^+$ analyzing ion beam was collimated using two sets of four-sector slits. The width $\times$ height of the beam spot was 0.2 mm $\times$ 1 mm. The beam divergence was kept below 0.06°. The beam current was measured using a transmission Faraday cup. The vacuum in the scattering chamber was kept at about $10^{-4}$ Pa. Hydrocarbon deposition was avoided by liquid $N_2$-cooled traps along the beam path and around the wall of the chamber. A Mylar foil with thickness of 6 μm was placed before the window of the ERDA detector to capture backscattered $He^+$ ions. Kapton foil as reference was used for calibration of the hydrogen content of the samples. ORTEC Si surface barrier detectors mounted at scattering angles of $\Theta = 165°$ and 20° were used to detect RBS and ERDA spectra, respectively. The detector resolution was about ~20 keV. The spectra were measured at sample tilt angles of 7° and 80° for RBS, and 80° for ERDA.

## 3. Results and Discussion

### 3.1. Optical Characterization of Thin Films

The refractive index and extinction coefficient are related to the interaction between a thin film and the incident light, indicating the optical properties of the thin films. SE is one of the most popular tools for characterizing the optical properties of different materials. The polarization state of the light changes while it is reflected or transmitted by the sample. Detection and interpretation of this change are the basis of the ellipsometric method. The thin films, reference R1 and thin films with hydrogen addition (S1–S6), were measured by SE in reflectance mode, where the complex reflectance ratio ($\bar{\rho}$) is recorded by the instrument. This quantity is usually expressed by the ψ and Δ ellipsometric angles in the following form:

$$\bar{\rho} = \frac{\bar{r}_p}{\bar{r}_s} = tg\psi \times e^{i\Delta} \tag{1}$$

where $\bar{\rho}$, $\bar{r}_p$, and $\bar{r}_s$ refer to the complex reflectance ratio, reflectance coefficient in p (parallel to the incident plane), and s (perpendicular to the incident plane) directions, respectively. The evaluation of the measurement data includes the modeling and fitting procedure. The differences between the modeled (generated) and real (measured) ψ and Δ ellipsometric angles are minimized by adjusting the free parameters of the applied model. The effects of the hydrogen flow applied to the different optical properties of the a-$SiN_x$:H thin films are shown in Figure 2. The refractive index of 1.96 is characteristic for the reference thin film (R1) sputtered in nitrogen atmosphere (Figure 2a). The addition of hydrogen until 3 sccm gives no or minimal effect on the refractive index. The increasing of hydrogen flow from 6 to 12 sccm results in a decrease in the refractive index from 1.96 to 1.89 (Figure 2a). We note that K. Mokkedem et al. reported a refractive index of 1.8 for hydrogenated silicon nitride thin films deposited under similar conditions using DC magnetron sputtering [16].

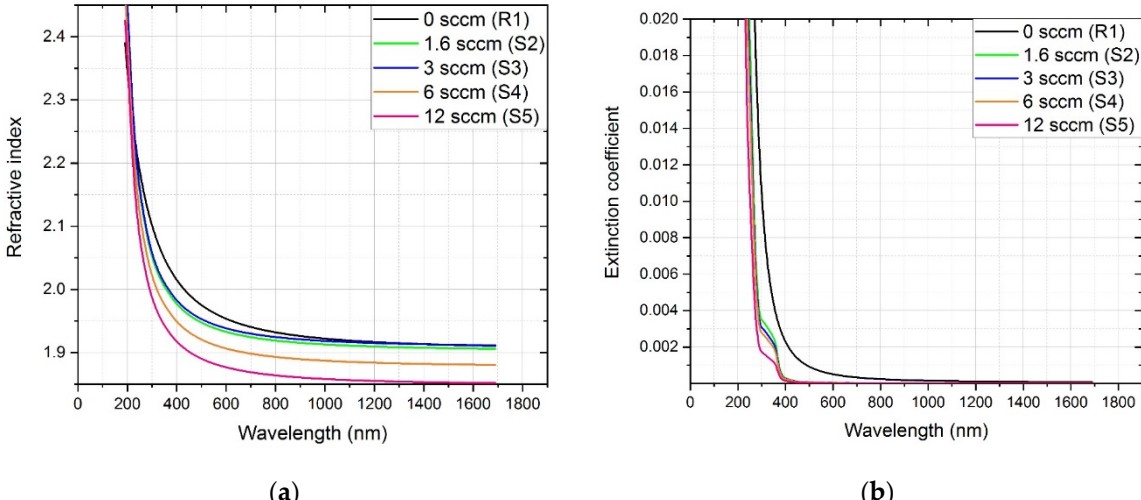

**Figure 2.** Effect of hydrogen flow on the optical properties of a-SiN$_x$ thin films: (**a**) refractive index, (**b**) extinction coefficient.

The refractive index of stoichiometric SiN is greatly dependent upon the deposition conditions, but it is greater than 2.0 at 630 nm [23]. The lower values of our RF-sputtered thin films indicate the presence of the non-stoichiometric SiN$_x$ phase. The extraction of the optical constants using the optical reflection spectra alone is usually very infrequent and complicated. The extinction coefficient values exhibit a large difference between the thin films grown with or without hydrogen (Figure 2b). There is a shift in the extinction coefficient from 400 to 300 nm, when hydrogen/nitrogen reactive sputtering is used.

The refractive index measured at 550 nm wavelength has a decreasing character as the partial pressure of hydrogen is increased during the sputtering process (Figure 3). K. Mokkedem et al. showed that the refractive index of their films vs. partial pressure of H$_2$ had a similar character. They confirmed the effect of the hydrogen partial pressure on the refractive index. These variations may be explained by hydrogen and nitrogen incorporation into the thin films [16].

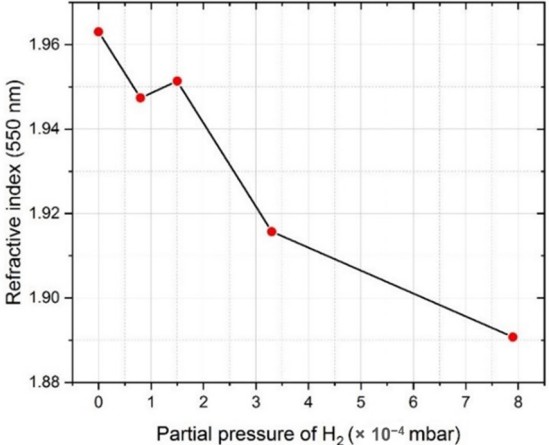

**Figure 3.** Refractive index of hydrogenated silicon nitride (SiN$_x$:H) thin films at a wavelength of 550 nm as a function of H$_2$ partial pressure.

### 3.2. Bonding Configuration and Chemical Composition

Understanding the chemical properties of silicon nitride and the role played by the hydrogen atoms that are incorporated during the growth of the film is a key factor for a-SiN$_x$:H applicability. FTIR is a widely used method for characterization of the bonding configuration in thin films. FTIR measurements allow to characterize the bonds involved in the material, and eventually to determine the bond densities and thus hydrogen concen-

tration thanks to the method developed by Lanford and Rand [24]. In the case of a-SiN$_x$:H, the presence of different hydrogen bonds can also be assumed based on the Lanford–Rand method [24].

Absorbance spectra of a-SiN$_x$ (R2) and a-SiN$_x$:H (S6) revealed the bonding configuration of the investigated thin films (Figure 4), consisting of typical absorption bands for different hydrogen bonding to nitrogen and silicon. The peak at 880 cm$^{-1}$ is associated with the Si–N stretching mode [25]. Peaks observed at 1175, 2200, and 3335 cm$^{-1}$ refer to the presence of the N–H bending mode, Si–H stretching mode, and N–H stretching mode, respectively [26].

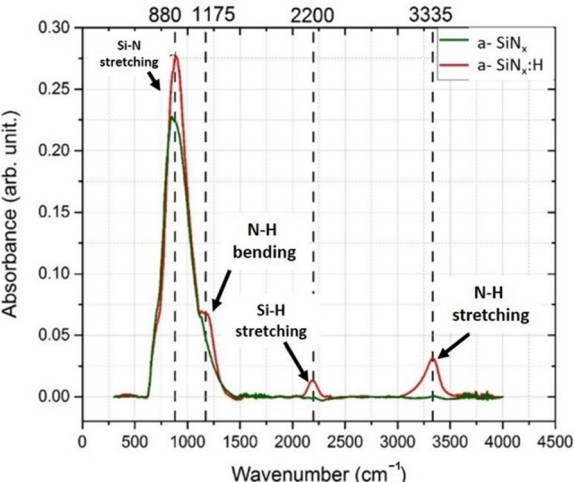

**Figure 4.** Absorbance spectra of reference a-SiN$_x$ (R2) and a-SiN$_x$:H (S6) thin films.

The concentrations of the N–H and Si–H bonds ($C_{Y-H}$) were calculated using the Lanford–Rand method [24] defined in the following form:

$$C_{Y-H} = \frac{A_{Y-H}}{ln10 \times \sigma_{Y-H}} = \frac{\int_{(Y-H)^s}^{(Y-H)^f} \alpha(\omega)d(\omega)}{ln10 \times \sigma_{Y-H}} \quad (2)$$

where Y corresponds to silicon (Si) or nitrogen (N) atoms, $A_{Y-H}$, $\sigma_{Y-H}$, $(Y-H)^s$, $(Y-H)^f$, $\alpha(\omega)$ are the concentration of Y–H bond in the cm$^{-3}$ unit, the normalized absorption area of the Y–H band, the absorption cross-section of the Y–H bond in the cm$^2$ unit, the beginning wavenumber of the Y–H band in the cm$^{-1}$ unit, the final wavelength of the Y–H band in the cm$^{-1}$ unit, and the spectral absorption coefficient, respectively.

The $\sigma_{N-H}$ and $\sigma_{Si-H}$ absorption cross-sections were determined by Lanford and Rand [24]: $\sigma_{N-H} = 5.3 \times 10^{-18}$ cm$^2$ and $\sigma_{Si-H} = 7.4 \times 10^{-18}$ cm$^2$. These values were used in Equation (2) above. Table 2 summarizes the parameters and the results of the calculations for both Si–H and N–H bond concentrations.

**Table 2.** Calculation parameters and results of the Si–H and N–H bond concentrations.

| Bond | $(Y–H)^s$ (cm$^{-1}$) | $(Y–H)^f$ (cm$^{-1}$) | $A_{Y-H}$ | $\sigma_{Y-H}$ ($10^{-18}$ cm$^{-2}$) | $C_{Y-H}$ ($10^{20}$ cm$^{-3}$) |
|------|------------|------------|------|------------|------------|
| N–H | 3026 | 3502 | 0.066 | 5.3 | 3.49 |
| Si–H | 1994 | 2299 | 0.016 | 7.4 | 0.61 |

If the hydrogen is one coordinated, then the total concentration of bonded hydrogen atoms in the film can be calculated by the sum of the concentration of N–H and Si–H bonds:

$$C_{H, bound} = C_{N-H} + C_{Si-H} \quad (3)$$

Based on Equation (3), the amount of total bonded hydrogen content of the S6 sample is $4.1 \times 10^{20}$ at/cm$^3$.

Rutherford backscattering spectrometry (RBS) in combination with elastic recoil detection analysis (ERDA) measurements were performed to obtain thin film elemental concentration depth profiles.

The atomic concentrations of silicon, nitrogen, free (unbounded), and bounded hydrogen as well as thin film atomic densities were determined (Table 3) from measured RBS/ERDA spectra, considering the layer thicknesses determined from TEM measurements. The samples show comparable atomic densities; only a-SiN$_x$:H sputtered with high hydrogen flow (S4) showed a lower value of $\sim 4.9 \times 10^{22}$ at/cm$^3$. In the case of the thin (S1) and thick a-SiN$_x$:H (S6) samples, the atomic density of measured hydrogen was found to be $0.46 \times 10^{22}$ and $0.59 \times 10^{22}$ at/cm$^3$, respectively (Table 3). We note that RBS/ERDA is not sensitive to the chemical bonding states and measures all the hydrogen content even if it is in atomic, bounded, or in H$_2$ molecular form.

**Table 3.** Atomic layer densities and concentrations (at.%) and atomic densities (at/cm$^3$) for the Si, N, and H components of a-SiN$_x$:H layers as evaluated from RBS/ERDA measurements using layer thicknesses (in nm) obtained from TEM analysis. * fully closed valve.

| H$_2$ Flow (sccm) | Atomic Layer Density ($10^{22}$ at/cm$^3$) | Silicon (Si) | | Nitrogen (N) | | Hydrogen (H) | |
|---|---|---|---|---|---|---|---|
| | | at.% | ($10^{22}$ at/cm$^3$) | at.% | ($10^{22}$ at/cm$^3$) | at.% | ($10^{22}$ at/cm$^3$) |
| 0 * | 6.8 | 43.2 | 3.1 | 50 | 3.59 | 6.8 | 0.46 |
| 0.9 | 6.25 | 40.4 | 2.48 | 52 | 3.19 | 7.6 | 0.47 |
| 1.6 | 6.6 | 43.3 | 2.73 | 45.3 | 2.85 | 11.4 | 0.75 |
| 3 | 4.9 | 35.2 | 1.72 | 52.4 | 2.57 | 12.4 | 0.61 |
| 12 | 6 | 33.1 | 1.99 | 57 | 3.42 | 9.9 | 0.59 |

Lanford et al. noted that the exact values of hydrogen content cannot be measured by FTIR, but a good approximation of the bonding structure is represented [24]. In contrast to effusion measurements, FTIR shows only the bonded amount of hydrogen in the material, whereas the effusion measurement detects the total hydrogen content including atomic hydrogen as well.

However, effusion of hydrogen is mostly detected as molecular H$_2$. In RF-sputtered a-SiN$_x$:H (S6), the concentration of bonded hydrogen calculated from FTIR spectra is only 4 at.%. The ERDA measurement confirmed 10 at.% of the total hydrogen concentration (Table 3). This means that 6 at.% hydrogen was incorporated during growth in molecular form. This fact is in good agreement with density measurements (Table 3). According to Dekkers et al., a low-density material induces the release of hydrogen in molecular form (H$_2$) and a denser SiN$_x$ makes the hydrogen desorption slower but its atomic form is preferred [27].

### 3.3. Activation Energy

Owing to the annealing chemical reactions such as breaking up and rebuilding of different bonds, diffusion of hydrogen atoms or molecules could take place. The energy must be provided for a certain chemical reaction (activation energy) of the process. As a result of chemical reactions, the annealing quality of the layer surface can be changed. The estimated activation energy can be detected by this change. The arrangement of the measurement described in Ref. [22] enables the determination of onset time of the layer surface change by monitoring the elapsed time and the alteration of the reflectivity of the sample surface.

The a-SiN$_x$:H thin film sputtered at a H$_2$ flow of 6 sccm (S4) was investigated at different temperatures. The temperature and the corresponding onset time values are summarized in Table 4. The Arrhenius equation [28] describes the rate constant of a

chemical reaction as a function of the temperature in terms of two empirical factors, namely $E_{exp}$ experimental activation energy and $A$ pre-exponential factor:

$$r(T) = Ae^{-\frac{E_{exp}}{k_B T}} \tag{4}$$

where $r$, $T$, and $k_B$ refer to the rate constant, temperature in $K$ units, and Boltzmann constant, respectively. $A$ and $E_{exp}$ can be determined from the parameters (increment and slope) of linear fit on $\ln(k)$ against $1/T$ experimental points.

**Table 4.** Measurement data of the a-SiN$_x$:H thin film sputtered at a H$_2$ flow of 6 sccm (S4) for the Arrhenius plot.

| Temperature (K) | Onset Time (s) |
|:---:|:---:|
| 320 | 292 |
| 323 | 149 |
| 326 | 57 |
| 330 | 28 |

The $r$ reaction rate constant against the inverse temperature of annealing and the linear fit on the data points are shown in Figure 5. The experimental activation energy ($E_{exp}$) deduced from the slope of the fitted line is $E_{exp} = 2.19 \pm 0.17$ eV. It must be noted that the error of $E_{exp}$ is much bigger than the error of the linear fit due to the uncertainty of time and temperature measurement.

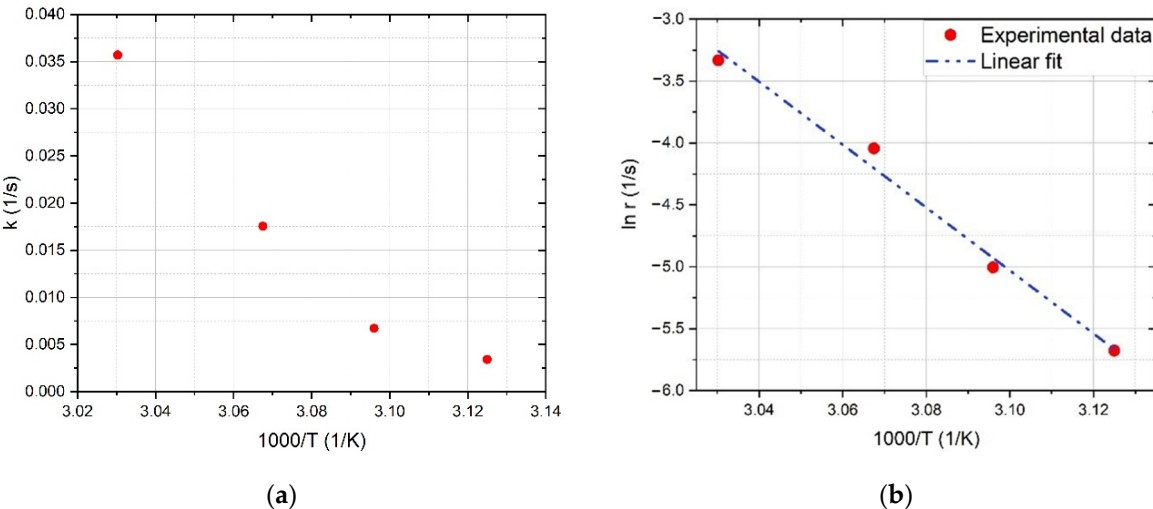

(a)    (b)

**Figure 5.** Arrhenius plot of the a-SiN$_x$:H thin film sputtered at a H$_2$ flow of 6 sccm (S4): (**a**) experimental data, (**b**) linear fit.

### 3.4. Structural and Morphological Characterization

Structural investigations confirmed the film thickness between ~142 nm and 155 nm (Figure 6). The selected area electron diffractions (SAEDs) proved that the deposited layer structure is amorphous without any crystalline structure (Figure 6). The SAED was provided by the ProcessDiffraction program [29–31].

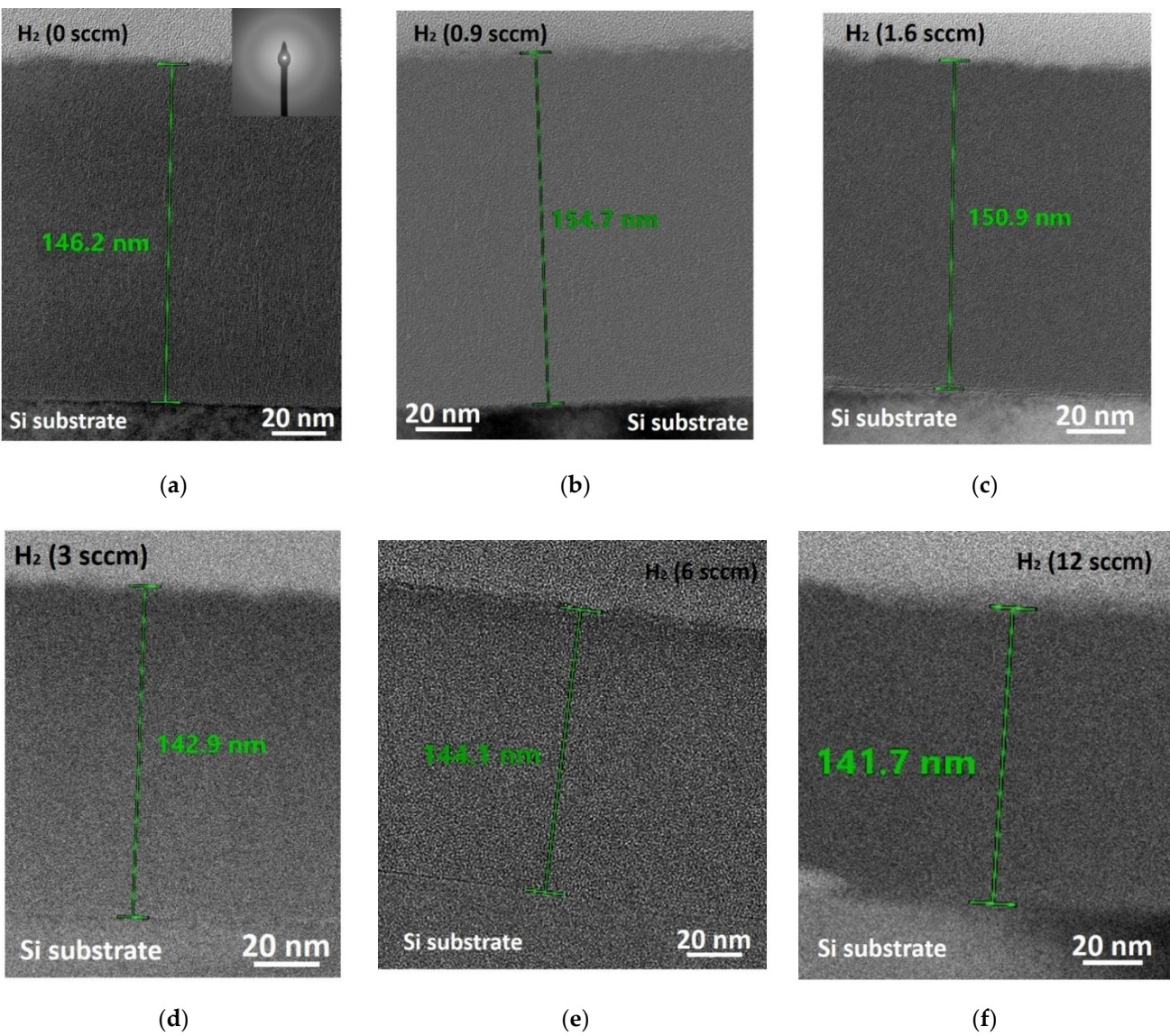

**Figure 6.** Cross-section TEM images of a-SiN and a-SiN$_x$:H thin films sputtered at different H$_2$ flow: (**a**) 0 sccm (R1) with selected area electron diffraction (SAED) detail, (**b**) 0.9 sccm (S1), (**c**) 1.6 sccm (S2), (**d**) 3 sccm (S3), (**e**) 6 sccm (S4), and (**f**)12 sccm (S5).

Such a large difference indicates a more complex incorporation mechanism of hydrogen in the silicon–nitrogen network and the presence of voids in a-SiN$_x$:H thin films. These structural observations correlated with the refractive index values. The refractive index decreased with increasing hydrogen flow and porosities during the growth process of the thin films. Similar results were obtained by AJ. Flewitt et al. when the nitrogen incorporation into a-Si:N:H films was caused by the dissociation of NH$_3$ molecules, leading to the reaction with a growing surface. This process resulted in a lower refractive index of the a-SiN$_x$:H [32].

TEM provides direct information on the structure of materials. Furthermore, the high-angle annular dark-field (HAADF) STEM allows the visibility of the porosities. The detailed study confirmed the dense thin film (Figure 7a) during hydrogen-free sputtering and the porous structure with nanometer-scale porosities homogenously distributed in the thin film sputtered at 1.6 sccm hydrogen flow (Figure 7b).

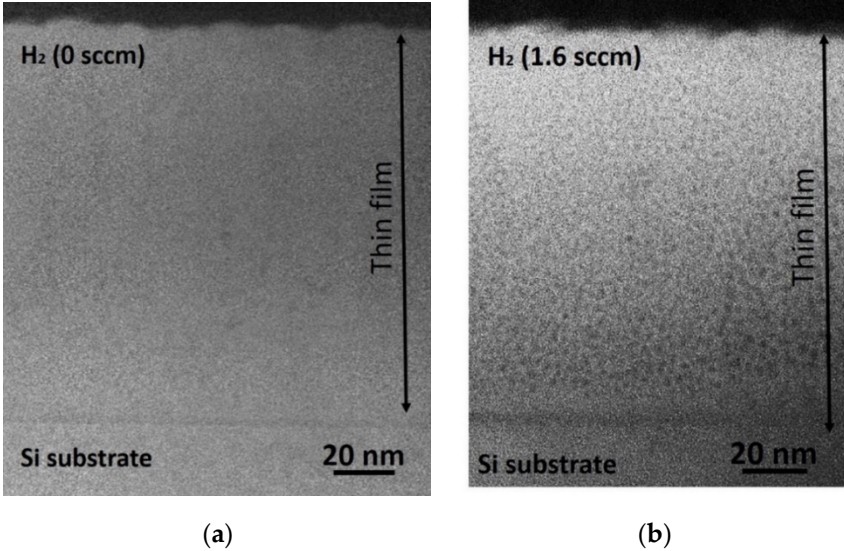

**(a)**           **(b)**

**Figure 7.** High-angle annular dark-field (HAADF) STEM images of a-SiN$_x$ thin films: (**a**) hydrogen-free a-SiN$_x$ thin film (S1), (**b**) a-SiN$_x$:H (S3).

The out-diffusion of hydrogen due to annealing plays a prominent role in the densification of thin films. The temperature of 800 °C was considered as a critical temperature, since the a-SiN$_x$ thin films are often subjected to thermal processes including rapid thermal annealing [33]. Morphological investigations of the thin film's surface before (Figure 8a) and after (Figure 8b) annealing show that the surface of a-SiN$_x$:H thin film changes due to heat treatment (Figure 8). The hydrogen, because of its molecular form, is released at a temperature of ~65 °C from the film. Blisters with a diameter of the order of 100 nm are created on the surface of the thin films. It is known from the literature [10,34,35] that, upon annealing, both hydrogen and nitrogen releases from a-SiN$_x$:H thin film were observed.

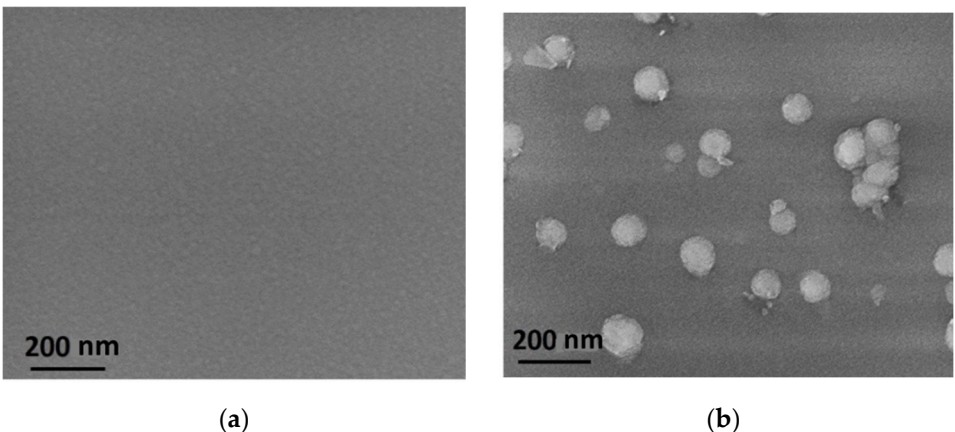

**(a)**           **(b)**

**Figure 8.** SEM images of a-SiN$_x$ sputtered at a H$_2$ flow rate of 12 sccm: (**a**) surface before annealing, (**b**) surface after annealing at 800 °C.

Similar to the effect reported in Ref. [22], blisters on the layer surface could be burst bubbles filled with hydrogen and/or nitrogen containing molecules. Initially, the volume of these bubbles was increasing due to thermal expansion of the fill-up gas during heat treatment. Then, at a critical point (at a given temperature after a certain time), the bubbles burst, leading to blister creation on the surface.

## 4. Conclusions

Amorphous hydrogenated silicon nitride thin films (a-SiN$_x$:H) have widespread applications from device passivation to light emitting diodes and antireflective coatings for solar cells. Chemical vapor deposition (CVD) and physical vapor deposition (PVD) are the most common techniques for silicon nitride film deposition with or without hydrogen addition. CVD-deposited film always contains hydrogen and its amount cannot be controlled directly during the preparation process. Due to this fact, the alternative fabrication method for controlled hydrogen concentration in a direct way from zero hydrogen content by adjusting the applied hydrogen gas flow to the chamber could be PVD (e.g., RF sputtering).

In this work, a-SiN$_x$:H films were sputtered at various H$_2$ flows with average thickness of 150 nm, and the effect of hydrogen incorporation on structural and optical properties was studied. The detailed structural characterization confirmed the formation of a dense thin film at hydrogen-free sputtering and a porous structure with homogenously distributed nanometer-scale porosities caused by hydrogen addition. The refractive index of 1.96 was characteristic for hydrogen-free SiN$_x$ thin films. Hydrogen flows up to 3 sccm were found to have no or minimal effect on the refractive index; for flows from 6 to 12 sccm, the refractive index decreased from 1.96 to 1.89, which can be explained by the hydrogen and nitrogen incorporation into the thin films. The calculations from FTIR spectra showed that a-SiN$_x$:H sputtered at 6 sccm H$_2$ flow presented a concentration of bounded hydrogen of ~4 at.%. The ERDA measurements confirmed a total hydrogen content of 10 at.%. This means that 6 at.% hydrogen was incorporated in a molecular form during the layer growth, which explained the lower density of the thin films. The out-diffusion of hydrogen due to annealing plays a prominent role in the densification of thin films. The molecular form of hydrogen is released at a temperature of ~65 °C from the film. Blisters with 100 nm diameter are created on the surface of the thin films. The low activation energy calculated by the Arrhenius method refers to significant diffusion of hydrogen molecules.

**Author Contributions:** N.H.: sample deposition, optical characterization, writing of the manuscript; R.L.: RF sputtering; M.S.: Arrhenius calculations; Z.Z.: ERDA measurements; P.P.: ellipsometry measurements; J.M.: FITR measurements; Z.F.: TEM measurements; C.B.: supervising, writing of the manuscript; K.B.: supervising, structural correlations, writing. All authors have read and agreed to the published version of the manuscript.

**Funding:** This research was funded by OTKA grant Nr. K131515, OTKA grant Nr. K131594, FLAG-ERA NKFIH 127723, NKFIH-NNE 129976.

**Institutional Review Board Statement:** Not applicable.

**Informed Consent Statement:** Not applicable.

**Data Availability Statement:** The data presented in this study are available on request from the corresponding author.

**Acknowledgments:** The authors would like to thank Levente Illés for SEM measurements and Andrea Fenyvesiné Jakab from the Centre for Energy Research for TEM sample preparations.

**Conflicts of Interest:** The authors declare no conflict of interest.

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
