# Peer review of "Examination of the Hydrogen Incorporation into Radio Frequency-Sputtered Hydrogenated SiNx Thin Films"

_coatings, doi:10.3390/coatings11010054_

Round 1

Reviewer 1 Report

Dear Author(s),

Thanks for the time for this monumental work. I have some questions regarding the manuscript. 

Line 95: What is the reason for choosing 2.10^5 mbar as the base pressure? Can the deposition chamber go higher pressure, such as 10^7 mbar?

Line 96; Why did N2 use instead of Ar?

Line 105; What is the angle between sample and target?

How did the authors measure the deposition rate? Have the authors used QCM to measure the thickness of deposition rate?

Line 282; What is the reason for ~150 nm thickness of the thin film? Can the authors go lower or higher?

The authors' used double side polished Si but could not see the comparison between single and double-sided samples in the results.

Table 1: How did the authors decide 30 min and 80 min deposition times?

Line 50; Can authors provide a reference(s) for the following sentence? 

"The CVD deposited film always contains hydrogen but its amount can't be controlled directly during the preparation process, only by several deposition parameters, such as the ratio of precursor gases or the substrate temperature. Due to this fact, magnetron sputtering technique could be the alternative fabrication method for a controlled hydrogen concentration in a direct way donw to zero by adjusting the applied hydrogen gas flow to the chamber."

It will be better if the authors add a figure to show the deposition process and system.

The authors need to check the grammar and spelling of this manuscript.

Thanks.

Author Response

Dear Reviewer, 

we would like to thank you for very detailed review and all suggestions. It will improve the final paper. We corrected our manuscript according to your comments. All corrections are listed in attached file.

Best regards,

Katalin Balázsi

Reviewer 2 Report

See attached file

Author Response

(The authors gave the same response as above.)

Reviewer 3 Report

This manuscript investigated the structural and optical properties of hydrogen-free and hydrogenated silicon nitride fabricated by radio frequency magnetron sputtering. I cannot recommend this manuscript for publication in its current form for the following reasons.

  1. The RFMS technique adopted in this manuscript and the fabricated silicon nitride films have been widely used and studied. The study in this manuscript does not emphasize its focus with sufficient clarity, i.e., why this fabrication technique of RFMS is used and what is the purpose of using characterization techniques such as SEM and TEM? In the results and discussion section, especially 3.5, there are few comparisons with the results of previous studies, and they all agree with the previous ones. So what is the purpose of this paper and how do the findings differ from previous studies? Which results are newly found and which concepts are newly proposed? These questions need to be answered clearly in the introduction and conclusion, from a more specific rather than general perspective.
  2. The plasma surface treatment process actually contains two separate parts, i.e., the effects of process parameters such as pressure, discharge power, etc. on the plasma discharge state, including various charged and active species densities and the electron temperature, and the effects of plasma parameters, such as the flux, energy, and angle of various species reaching the substrate, on the properties of deposited thin films. In the absence of plasma diagnostic equipment, plasma parameters can be left unstudied. However, please remind readers in the manuscript that the film properties are not directly determined by the process parameters, and the discharge states obtained by different devices at the same process parameters may be completely different and only serve as a reference when plasma diagnostic results are not available.
  3. There are a lot of typos and grammatical errors in the manuscript, e.g., line 16, "Frequecy" should be "Frequency"; line 21, "refracting index" should be "refractive index"; line 24 and 295, "diamater" should be "diameter"; line 53, "donw" should be "down"; line 78, "modell" should be "model"; line 163, "extintion" should be "extinction"; line 165, a redundant "the"; line 248, "porose" should be "porous"; and many, many more. Please read the manuscript carefully and revise it carefully throughout.
  4. There are other formatting errors. The application of punctuation in the abstract is really strange, with all commas replaced by semicolons and a missing period at the end of the abstract. Equation (2) is incomplete. The text markings in Figure 5(a), (d), (e), and (f) are incorrect. The format of the final reference section is not consistent.

Author Response

Dear Reviewer,

we would like to thank you for very detailed review and all suggestions. It will improve the final paper. We corrected our manuscript according to your comments. All corrections are listed in attached file

Round 2

Reviewer 1 Report

Thanks to authors for the update.  I don't have more questions. 

Author Response

Dear Reviewer,

thank you for your acceptance.

Best regards,

Katalin Balazsi

Reviewer 3 Report

This manuscript has been revised in accordance with the referees' suggestions to a point that I can recommend it for publication in Coatings.

Author Response

(The authors gave the same response as above.)
